# Peer review of "Morphology Generation for Statistical Machine Translation using Deep Learning Techniques"

_ACL 2017 — decision unknown_

[Official Review · Reviewer 1 · rating 1 · confidence 4]
soundness 5 · originality 5 · clarity 3 · impact 3 · substance 4 · appropriateness 5 · meaningful comparison 3 · presentation format Oral Presentation

This paper details a method of achieving translation from morphologically
impoverished languages (e.g. Chinese) to morphologically rich ones (e.g.
Spanish) in a two-step process. First, a system translates into a simplified
version of the target language. Second, a system chooses morphological features
for each generated target word, and inflects the words based on those features.

While I wish the authors would apply the work to more than one language pair, I
believe the issue addressed by this work is one of the most important and
under-addressed problems with current MT systems. The approach taken by the
authors is very different than many modern approaches based on BPE and
character-level models, and instead harkens back to approaches such as
"Factored Translation Models" (Koehn and Hoang, 2007) and "Translating into
Morphologically Rich Languages with Synthetic Phrases" (Chahuneau et a. 2013),
both of which are unfortunately uncited.

I am also rather suspicious of the fact that the authors present only METEOR
results and no BLEU or qualitative improvements. If BLEU scores do not rise,
perhaps the authors could argue why they believe their approach is still a net
plus, and back the claim up with METEOR and example sentences.

Furthermore, the authors repeatedly talk about gender and number as the two
linguistic features they seek to correctly handle, but seem to completely
overlook person. Perhaps this is because first and second person pronouns and
verbs rarely occur in news, but certainly this point at least merits brief
discussion. I would also like to see some discussion of why rescoring hurts
with gender. If the accuracy is very good, shouldn the reranker learn to just
keep the 1-best?

Finally, while the content of this paper is good overall, it has a huge amount
of spelling, grammar, word choice, and style errors that render it unfit for
publication in its current form. Below is dump of some errors that I found.

Overall, I would like to this work in a future conference, hopefully with more
than one language pair, more evaluation metrics, and after further
proofreading.

General error dump:
Line 062: Zhand --> Zhang
Line 122: CFR --> CRF
Whole related work section: consistent use of \cite when \newcite is
appropriate
It feels like there's a lot of filler: "it is important to mention that", "it
is worth mentioning that", etc
Line 182, 184: "The popular phrase-based MT system" = moses? or PBMT systems in
general?
Line 191: "a software"
Line 196: "academic and commercial level" -- this should definitely be
pluralized, but are these even levels?
Line 210: "a morphology-based simplified target" makes it sound like this
simplified target uses morphology. Perhaps the authors mean "a morphologically
simplified target"?
Line 217: "decide on the morphological simplifications"?
Table 1: extra space in "cuestión" on the first line and "titulado" in the
last line.
Table 1: Perhaps highlight differences between lines in this table somehow?
How is the simplification carried out? Is this simplifier hand written by the
authors, or does it use an existing tool?
Line 290: i.e. --> e.g.
Line 294: "train on" or "train for"
Line 320: "our architecture is inspired by" or "Collobert's proposal inspires
our architecture"
Line 324: drop this comma
Line 338: This equation makes it look like all words share the same word vector
W
Line 422: This could also be "casas blancas", right? How does the system choose
between the sg. and pl. forms? Remind the reader of the source side
conditioning here.
Line 445: This graph is just a lattice, or perhaps more specifically a "sausage
lattice"
Line 499: Insert "e.g." or similiar: (e.g. producirse)
Line 500: misspelled "syllable"
Line 500/503: I'd like some examples or further clarity on what palabras llanas
and palabras estrújulas are and how you handle all three of these special
cases.
Line 570: "and sentences longer than 50 words"
Line 571: "by means of zh-seg" (no determiner) or "by means of the zh-seg tool"
Line 574: are you sure this is an "and" and not an "or"?
Line 596: "trained for" instead of "trained on"
Line 597: corpus --> copora
Line 604: size is --> sizes are
Line 613: would bigger embedding sizes help? 1h and 12h are hardly unreasonable
training times.
Line 615: "seven and five being the best values"
Line 617: Why 70? Increased from what to 70?
Table 3: These are hyperparameters and not just ordinary parameters of the
model
Line 650: "coverage exceeds 99%"?
Line 653: "descending"
Line 666: "quadratic"
Line 668: space before \cites
Line 676: "by far" or "by a large margin" instead of "by large"
Line 716: below
Line 729: "The standard phrase-based ..."
zh-seg citation lists the year as 2016, but the tool actually was released in
2009

[Official Review · Reviewer 2 · rating 2 · confidence 4]
soundness 5 · originality 5 · clarity 4 · impact 3 · substance 4 · appropriateness 5 · meaningful comparison 3 · presentation format Poster

The paper describes a method for improving two-step translation using deep
learning. Results are presented for Chinese->Spanish translation, but the
approach seems to be largely language-independent.

The setting is fairly typical for two-step MT. The first step translates into a
morphologically underspecified version of the target language. The second step
then uses machine learning to fill in the missing morphological categories and
produces the final system output by inflecting the underspecified forms (using
a morphological generator). The main novelty of this work is the choice of deep
NNs as classifiers in the second step. The authors also propose a rescoring
step which uses a LM to select the best variant.

Overall, this is solid work with good empirical results: the classifier models
reach a high accuracy (clearly outperforming baselines such as SVMs) and the
improvement is apparent even in the final translation quality.

My main problem with the paper is the lack of a comparison with some
straightforward deep-learning baselines. Specifically, you have a structured
prediction problem and you address it with independent local decisions followed
by a rescoring step. (Unless I misunderstood the approach.) But this is a
sequence labeling task which RNNs are well suited for. How would e.g. a
bidirectional LSTM network do when trained and used in the standard sequence
labeling setting? After reading the author response, I still think that
baselines (including the standard LSTM) are run in the same framework, i.e.
independently for each local label. If that's not the case, it should have been
clarified better in the response. This is a problem because you're not using
the RNNs in the standard way and yet you don't justify why your way is better
or compare the two approaches.

The final re-scoring step is not entirely clear to me. Do you rescore n-best
sentences? What features do you use? Or are you searching a weighted graph for
the single optimal path? This needs to be explained more clearly in the paper.
(My current impression is that you produce a graph, then look for K best paths
in it, generate the inflected sentences from these K paths and *then* use a LM
-- and nothing else -- to select the best variant. But I'm not sure from
reading the paper.) This was not addressed in the response.

You report that larger word embeddings lead to a longer training time. Do they
also influence the final results?

Can you attempt to explain why adding information from the source sentence
hurts? This seems a bit counter-intuitive -- does e.g. the number information
not get entirely lost sometimes because of this? I would appreciate a more
thorough discussion on this in the final version, perhaps with a couple of
convincing examples.

The paper contains a number of typos and the general level of English may not
be sufficient for presentation at ACL.

Minor corrections:

context of the application of MT -> context of application for MT

In this cases, MT is faced in two-steps -> In this case, MT is divided into two
steps

markov -> Markov

CFR -> CRF

task was based on a direct translation -> task was based on direct translation

task provided corpus -> task provided corpora

the phrase-based system has dramatically -> the phrase-based approach...

investigated different set of features -> ...sets of features

words as source of information -> words as the source...

correspondant -> corresponding

Classes for gender classifier -> Classes for the...

for number classifier -> for the...

This layer's input consists in -> ...consists of

to extract most relevant -> ...the most...

Sigmoid does not output results in [-1, 1] but rather (0, 1). A tanh layer
would produce (-1, 1).

information of a word consists in itself -> ...of itself

this $A$ set -> the set $A$

empty sentences and longer than 50 words -> empty sentences and sentences
longer than...

classifier is trained on -> classifier is trained in

aproximately -> approximately

coverage raises the 99% -> coverage exceeds 99% (unless I misunderstand)

in descendant order -> in descending order

cuadratic -> quadratic (in multiple places)

but best results -> but the best results

Rescoring step improves -> The rescoring step...

are not be comparable -> are not comparable

[Official Review · Reviewer 3 · rating 2 · confidence 5]
soundness 5 · originality 5 · clarity 3 · impact 3 · substance 4 · appropriateness 5 · meaningful comparison 3 · presentation format Poster

This paper presents a method for generating morphology, focusing on gender and
number, using deep learning techniques. From a morphologically simplified
Spanish text, the proposed approach uses a classifier to reassign the gender
and number for each token, when necessary. The authors compared their approach
with other learning algorithms, and evaluated it in machine translation on the
Chinese-to-Spanish (Zh->Es) translation direction.

Recently, the task of generating gender and number has been rarely tackled,
morphology generation methods usually target, and are evaluated on,
morphologically-rich languages like German or Finnish.
However, calling the work presented in this paper “morphology
generation“ is a bit overselling as the proposed method clearly deals only
with
gender and number. And given the fact that some rules are handcrafted for this
specific task, I do not think this method can be straightforwardly applied to
do more complex morphology generation for morphologically-rich languages.

This paper is relatively clear in the sections presenting the proposed method.
A
lot of work has been done to design the method and I think it can have some
interesting impact on various NLP tasks. However the evaluation part of
this work is barely understandable as many details of what is done, or why it
is done, are missing. From this evaluation, we cannot know if the proposed
method brings improvements over state-of-the-art methods while the experiments
cannot be replicated. Furthermore, no analysis of the results obtained is
provided. Since half a page is still available, there was the possibility
to provide more information to make more clear the evaluation. This work lacks
of motivation. Why do you think deep learning can especially improve gender and
number generation over state-of-the-art methods?

In your paper, the word “contribution“ should be used more wisely, as it is
now in the paper, it is not obvious what are the real contributions (more
details below). 

abstract:
what do you mean by unbalanced languages?

section 1:
You claim that your main contribution is the use of deep learning. Just the use
of deep learning in some NLP task is not a contribution.

section 2:
You claim that neural machine translation (NMT), mentioned as “neural
approximations“,  does not achieve state-of-the-art results for Zh->Es. I
recommend to remove this claim from the paper, or to discuss it more, since
Junczys-Dowmunt et al. (2016), during the last IWSLT, presented some results
for Zh->Es with the UN corpus, showing that NMT outperforms SMT by around 10
BLEU points.

section 5.1:
You wrote that using the Zh->Es language pair is one of your main
contributions. Just using a language pair is not a contribution. Nonetheless, I
think it is nice to see a paper on machine translation that does not focus of
improving machine translation for English.
The numbers provided in Table 2 were computed before or after preprocessing?
Why did you remove the sentences longer than 50 tokens?
Precise how did you obtain development and test sets, or provide them. Your
experiments are currently no replicable especially because of that.

section 5.2:
You wrote that you used Moses and its default parameters, but the default
parameters of Moses are not the same depending on the version, so you should
provide the number of the version used.

section 5.3:
What do you mean by “hardware cost“?
Table 3: more details should be provided regarding how did you obtain these
values. You chose these values given the classifier accuracy, but how precisely
and on what data did you train and test the classifiers? On the same data used
in section 6?
If I understood the experiments properly, you used simplified Spanish. But I
cannot find in the text how do you simplify Spanish. And how do you use it to
train the classifier and the SMT system? 

section 6:
Your method is better than other classification
algorithms, but it says nothing about how it performs compared to the
state-of-the-art methods. You should at least precise why you chose these
classifications algorithms for comparison. Furthermore, how your rules impact
these results? And more generally, how do you explain such a high accuracy for
you method?
Did you implement all these classification algorithms by yourselves? If not,
you must provide the URL or cite the framework you used.
For the SMT experiments, I guess you trained your phrase table on simplified
Spanish. You must precise it.
You chose METEOR over other metrics like BLEU to evaluate your results. You
must provide some explanation for this choice. I particularly appreciate when I
see a MT paper that does not use BLEU for evaluation, but if you use METEOR,
you must mention which version you used. METEOR has largely changed since 2005.
You cited the paper of 2005, did you use the 2005 version? Or did you use the
last one with paraphrases? 
Are your METEOR scores statistically significant?

section 7:
As future work you mentioned “further simplify morphology“. In this paper,
you do not present any simplification of morphology, so I think that choosing
the word
“further“ is misleading.

some typos:
femenine
ensambling
cuadratic

style:
plain text citations should be rewritten like this: “(Toutanova et al, 2008)
built “ should be “Toutanova et al. (2008) built “
place the caption of your tables below the table and not above, and with more
space between the table and its caption.
You used the ACL 2016 template. You must use the new one prepared for ACL 2017.
More generally, I suggest that you read again the FAQ and the submission
instructions provided on the ACL 2017 website. It will greatly help you to
improve the paper. There are also important information regarding references:
you must provide DOI or URL of all ACL papers in your references.

-----------------------

After authors response:

Thank you for your response.

You wrote that rules are added just as post-processing, but does it mean that
you do not apply them to compute your classification results? Or if you do
apply them before computing these results, I'm still wondering about their
impact on these results.

You wrote that Spanish is simplified as shown in Table 1, but it does not
answer my question: how did you obtain these simplifications exactly? (rules?
software? etc.) The reader need to now that to reproduce your approach.

The classification algorithms presented in Table 5 are not state-of-the-art, or
if they are you need to cite some paper. Furthermore, this table only tells
that deep learning gives the best results for classification, but it does not
tell at all if your approach is better than state-of-the-art approach for
machine translation. You need to compare your approach with other
state-of-the-art morphology generation approaches (described in related work)
designed for machine translation. If you do that your paper will be much more
convincing in my opinion.